# A Versatile Integral in Physics and Astronomy and Fox's H-Function

**Arak M. Mathai** [1] **and Hans J. Haubold** [2,*] 

[1] Department of Mathematics and Statistics, McGill University, Montreal, PQ H3A 2K6, Canada; a.mathai@mcgill.ca
[2] Vienna International Centre, A-1400 Vienna, Austria
\* Correspondence: hans.haubold@gmail.com

**Abstract:** This paper deals with a general class of integrals, the particular cases of which are connected to outstanding problems in physics and astronomy. Nuclear reaction rate probability integrals in nuclear physics, Krätzel integrals in applied mathematical analysis, inverse Gaussian distributions, generalized type-1, type-2, and gamma families of distributions in statistical distribution theory, Tsallis statistics and Beck–Cohen superstatistics in statistical mechanics, and Mathai's pathway model are all shown to be connected to the integral under consideration. Representations of the integral in terms of Fox's H-function are pointed out.

**Keywords:** H-function; generalized beta; gamma; inverse Gaussian densities; pathway model

---

## 1. The Integral and the H-Function

In this paper we will consider a general class of integrals connected with the pathway model of [1]. These will enable us to address a wide range of problems in different areas such as inverse Gaussian processes in the area of stochastic processes, Krätzel integrals in applied analysis, generalized type-1, type-2, and gamma densities in statistical distribution theory, Tsallis non-extensive statistical mechanics, Beck–Cohen superstatistics in astrophysics problems, reaction probability integrals in nuclear physics, and other related problems, which may be seen from the formalism introduced in this paper. For the extension of this integral to incorporate quantum-tail effects see [2]. Consider the following integral:

$$f(z_2|z_1) = \int_0^\infty x^{\gamma-1}[1 + z_1^\delta(\alpha-1)x^\delta]^{-\frac{1}{\alpha-1}}[1 + z_2^\rho(\beta-1)x^{-\rho}]^{-\frac{1}{\beta-1}}\mathrm{d}x \tag{1}$$

$$\text{for } \alpha > 1, \beta > 1, z_1 \geq 0, z_2 \geq 0, \delta > 0, \rho > 0, \Re(\gamma+1) > 0,$$

$$\Re(\frac{1}{\alpha-1} - \frac{\gamma+1}{\delta}) > 0, \Re(\frac{1}{\beta-1} - \frac{1}{\rho}) > 0$$

$$= \int_0^\infty \frac{1}{x} f_1(x) f_2(\frac{z_2}{x})\mathrm{d}x \tag{2}$$

where $\Re(\cdot)$ denotes the real part of $(\cdot)$,

$$f_1(x) = x^\gamma[1 + z_1^\delta(\alpha-1)x^\delta]^{-\frac{1}{\alpha-1}}, \ f_2(x) = [1 + (\beta-1)x^\rho]^{-\frac{1}{\beta-1}} \tag{3}$$

with Mellin transforms

$$M_{f_1}(s) = [\delta z_1^{\gamma+s}(\alpha-1)^{\frac{\gamma+s}{\delta}}]^{-1}\frac{\Gamma(\frac{\gamma+s}{\delta})\Gamma(\frac{1}{\alpha-1}-\frac{\gamma+s}{\delta})}{\Gamma(\frac{1}{\alpha-1})}, \tag{4}$$

$$\Re(\gamma+s) > 0, \Re\left(\frac{1}{\alpha-1}-\frac{\gamma+s}{\delta}\right) > 0$$

and

$$M_{f_2}(s) = [\rho(\beta-1)^{\frac{s}{\rho}}]^{-1}\frac{\Gamma(\frac{s}{\rho})\Gamma(\frac{1}{\beta-1}-\frac{s}{\rho})}{\Gamma(\frac{1}{\beta-1})} \tag{5}$$

$$\Re(s) > 0, \Re\left(\frac{1}{\alpha-1}-\frac{s}{\rho}\right) > 0.$$

Hence, the Mellin transform of $f(z_2|z_1)$, as a function of $z_2$, with parameter $s$ is the following:

$$\begin{aligned}
M_{f(z_2|z_1)}(s) &= M_{f_1}(s)M_{f_2}(s) \\
&= \frac{1}{\delta\,z_1^{\gamma+s}(\alpha-1)^{\frac{\gamma+s}{\delta}}}\frac{\Gamma(\frac{\gamma+s}{\delta})\Gamma(\frac{1}{\alpha-1}-\frac{\gamma+s}{\delta})}{\Gamma(\frac{1}{\alpha-1})} \\
&\times \frac{1}{\rho(\beta-1)^{\frac{s}{\rho}}}\frac{\Gamma(\frac{s}{\rho})\Gamma(\frac{1}{\beta-1}-\frac{s}{\rho})}{\Gamma(\frac{1}{\beta-1})} \\
&\text{for } \Re(\gamma+s) > 0, \Re(\frac{1}{\alpha-1}-\frac{\gamma+s}{\delta}) > 0, \Re(s) > 0, \\
&\Re(\frac{1}{\beta-1}-\frac{s}{\rho}) > 0, z_1 > 0, z_2 > 0.
\end{aligned} \tag{6}$$

Putting $y = \frac{1}{x}$ in (1) we have

$$f(z_1|z_2) = \int_0^\infty \frac{y^{-\gamma}}{y}[1+z_1^\delta(\alpha-1)y^{-\delta}]^{-\frac{1}{\alpha-1}}[1+z_2^\rho(\beta-1)y^\rho]^{-\frac{1}{\beta-1}}\,dy. \tag{7}$$

Evaluating the Mellin transform of (7) with parameter $s$ and treating it as a function of $z_1$, we have exactly the same expression in (6). Hence

$$M_{f(z_2|z_1)}(s) = M_{f(z_1|z_2)}(s) = \text{right side in (6).} \tag{8}$$

By taking the inverse Mellin transform of $M_{f(z_2|z_1)}(s)$ one can get the integral $f(z_2|z_1)$ as an H-function as follows:

$$f(z_2|z_1) = c^{-1}H_{2,2}^{2,2}\left[z_1z_2(\alpha-1)^{\frac{1}{\delta}}(\beta-1)^{\frac{1}{\rho}}\Big|_{(\frac{\gamma}{\delta},\frac{1}{\delta}),(0,\frac{1}{\rho})}^{(1-\frac{1}{\alpha-1}+\frac{\gamma}{\delta},\frac{1}{\delta}),(1-\frac{1}{\beta-1},\frac{1}{\rho})}\right], \tag{9}$$

where

$$c = \delta\rho z_1^\gamma(\alpha-1)^{\frac{\gamma}{\delta}}\Gamma(\frac{1}{\alpha-1})\Gamma(\frac{1}{\beta-1}),$$

and where $H_{p,q}^{m,n}(\cdot)$ is a H-function which is defined as the following Mellin–Barnes integral:

$$H_{p,q}^{m,n}\left[z\Big|_{(b_1,\beta_1),\dots,(b_q,\beta_q)}^{(a_1,\alpha_1),\dots,(a_p,\alpha_p)}\right] = \frac{1}{2\pi i}\int_L \phi(s)\,ds, \tag{10}$$

where

$$\phi(s) = \frac{\{\prod_{j=1}^{m} \Gamma(b_j + \beta_j s)\}\{\prod_{j=1}^{n} \Gamma(1 - a_j - \alpha_j s)\}}{\{\prod_{j=m+1}^{q} \Gamma(1 - b_j - \beta_j s)\}\{\prod_{j=n+1}^{p} \Gamma(a_j + \alpha_j s)\}}, \tag{11}$$

where $L$ is a suitable contour, $\alpha_j, j = 1, \ldots p, \beta_j, j = 1, \ldots, q$ are real positive numbers, $b_j, j = 1, \ldots, q, a_j, j = 1, \ldots, p$ are complex numbers and $L$ separates the poles of $\Gamma(b_j + \beta_j s), j = 1, \ldots m$ from those of $\Gamma(1 - a_j - \alpha_j s), j = 1, \ldots, n$. For more details about the theory and applications of the H-function [3].

The integral in (1) is connected to the reaction rate probability integral in nuclear reaction rate theory in the non-resonant case, Tsallis statistics in non-extensive statistical mechanics, superstatistics in astrophysics, generalized type-2, type-1 beta and gamma families of densities and the density of a product of two real positive random variables in statistical literature, Krätzel integrals in applied analysis, inverse Gaussian distribution in stochastic processes and other applications.

Observe that $f_1(x)$ and $f_2(x)$ in (3), multiplied by the appropriate normalizing constants can produce statistical densities. Further, $f_1(x)$ and $f_2(x)$ are defined for $-\infty < \alpha < \infty, -\infty < \beta < \infty$. When $\alpha > 1$ and $z_1 > 0, \delta > 0, f_1(x)$ multiplied by the normalizing constant stays in the generalized type-2 beta family. When $\alpha < 1$, writing $\alpha - 1 = -(1 - \alpha), \alpha < 1$ the function $f_1(x)$ switches into a generalized type-1 beta family and when $\alpha \to 1$,

$$\lim_{\alpha \to 1} f_1(x) = e^{-z_1^\delta x^\delta} \tag{12}$$

and hence $f_1(x)$ goes into a generalized gamma family. Similar is the behavior of $f_2(x)$ when $\beta$ ranges from $-\infty$ to $\infty$. Thus, the parameters $\alpha$ and $\beta$ create pathways to switch into different functional forms or different families of functions. Hence, we will call $\alpha$ and $\beta$ pathway parameters in this case. Let us look into some interesting special cases. Take the special case $\beta \to 1$,

$$f_1(z_2|z_1) = \int_0^\infty x^{\gamma-1}[1 + z_1^\delta(\alpha - 1)x^\delta]^{-\frac{1}{\alpha-1}} e^{-z_2^\rho x^{-\rho}} dx \tag{13}$$

$$\alpha > 1, z_1 > 0, z_2 > 0, \delta > 0, \rho > 0. \tag{14}$$

Put $y = \frac{1}{x}$

$$f_1(z_1|z_2) = \int_0^\infty y^{-\gamma-1}[1 + z_1^\delta(\alpha - 1)y^{-\delta}]^{-\frac{1}{\alpha-1}} e^{-z_2^\rho y^\rho} dy \tag{15}$$

$$\alpha > 1, z_1 > 0, z_2 > 0, \delta > 0, \rho > 0. \tag{16}$$

Let $\alpha \to 1$ in (1)

$$f_2(z_2|z_1) = \int_0^\infty x^{\gamma-1} e^{-z_1^\delta x^\delta}[1 + z_2^\rho(\beta - 1)x^{-\rho}]^{-\frac{1}{\beta-1}} dx \tag{17}$$

$$\beta > 1, z_1 > 0, z_2 > 0, \delta > 0, \rho > 0. \tag{18}$$

$$f_2(z_1|z_2) = \int_0^\infty x^{-\gamma-1} e^{-z_1^\delta x^{-\delta}}[1 + z_2^\rho(\beta - 1)x^\rho]^{-\frac{1}{\beta-1}} dx \tag{18}$$

$$\beta > 1, z_1 > 0, z_2 > 0, \delta > 0, \rho > 0. \tag{19}$$

Take $\alpha \to 1, \beta \to 1$ in (1)

$$f_3(z_2|z_1) = \int_0^\infty x^{\gamma-1} e^{-z_1^\delta x^\delta - z_2^\rho x^{-\rho}} dx \tag{20}$$

$$z_1 > 0, z_2 > 0, \delta > 0, \rho > 0.$$

$$f_3(z_1|z_2) = \int_0^\infty x^{-\gamma-1} e^{-z_1^\delta x^{-\delta} - z_2^\rho x^\rho} dx \tag{21}$$

$$z_1 > 0, z_2 > 0, \delta > 0, \rho > 0.$$

In all the integrals considered so far, we had one pathway factor containing $x^\delta$ and another pathway factor containing $x^{-\rho}$, where both the parameters $\delta > 0$ and $\rho > 0$, in the integrand. Also, the

integrand consisted of non-negative integrable functions and hence one could make statistical densities out of them. In statistical terms, all the integrals discussed so far will correspond to the density of $u = x_1 x_2$, where $x_1$ and $x_2$ are real scalar random variables, which are statistically independently distributed. Also, they fall in the category of Mellin convolution of a product involving two functions.

Now we will consider a class of integrals where the integrand consists of two pathway factors where both contain powers of $x$ of the form $x^\delta$ and $x^\rho$ with both $\delta$ and $\rho$ positive. Such integrals will lead to integrals of the following forms in the limits when the pathway parameters $\alpha$ and $\beta$ go to 1:

$$\int_0^\infty x^\gamma e^{-ax^\delta - bx^\rho} dx,$$

$a > 0, b > 0, \delta > 0, \rho > 0$. Observe that the evaluation of such an integral provides a method of evaluating Laplace transform of generalized gamma densities by taking one of the exponents $\delta$ or $\rho$ as unity. Consider the integral

$$I_4 = \int_0^\infty x^\gamma [1 + z_1^\delta(\alpha - 1)x^\delta]^{-\frac{1}{\alpha-1}} [1 + z_2^\rho(\beta - 1)x^\rho]^{-\frac{1}{\beta-1}} dx, \tag{22}$$

$\alpha > 1, \beta > 1, z_1 > 0, z_2 > 0, \delta > 0, \rho > 0$. Since the integrand consists of positive integrable functions, from a statistical point of view, the integral $I_4$ can be looked upon as the density of $u = \frac{x_1}{x_2}$, where $x_1$ and $x_2$ are real scalar random variables which are independently distributed or it can be looked upon as a convolution integral of the type

$$\int_0^\infty v f_1(uv) f_2(v) dv. \tag{23}$$

Let us take

$$f_1(x_1) = c_1 [1 + (\alpha - 1)x_1^\delta]^{-\frac{1}{\alpha-1}}, u = z_1$$
$$f_2(x_2) = c_2 x^{\gamma-1}[1 + z_2^\rho(\beta - 1)x_2^\rho]^{-\frac{1}{\beta-1}}$$

Taking the Mellin transforms and writing as expected values, where $E(\cdot)$ denotes the expected value of $(\cdot)$

$$E(x_1)^{s-1} = c_1 \int_0^\infty x_1^{s-1}[1 + (\alpha - 1)x_1^\delta]^{-\frac{1}{\alpha-1}} dx_1$$

$$= \frac{c_1}{\delta(\alpha-1)^{\frac{s}{\delta}}} \frac{\Gamma\left(\frac{s}{\delta}\right)\Gamma\left(\frac{1}{\alpha-1} - \frac{s}{\delta}\right)}{\Gamma\left(\frac{1}{\alpha-1}\right)}, \Re(s) > 0, \Re(\frac{1}{\alpha-1} - \frac{s}{\delta}) > 0,$$

$$E(x_2^{1-s}) = c_2 \int_0^\infty x_2^{\gamma-s}[1 + z_2^\rho(\beta - 1)x_2^\rho]^{-\frac{1}{\beta-1}} dx_2$$

$$= \frac{c_2}{\rho[z_2^\rho(\beta-1)]^{\frac{\gamma-s+1}{\rho}}} \frac{\Gamma\left(\frac{\gamma-s+1}{\rho}\right)\Gamma\left(\frac{1}{\beta-1} - \frac{\gamma-s+1}{\rho}\right)}{\Gamma\left(\frac{1}{\beta-1}\right)}$$

$$\Re(\gamma - s + 1) > 0, \Re(\frac{1}{\beta-1} - \frac{\gamma-s+1}{\rho}) > 0.$$

Therefore the density of $u = \frac{x_1}{x_2}$ is given by

$$g(u) = \frac{c_1 c_2}{\delta\rho[z_2^\rho(\beta-1)]^{\frac{\gamma+1}{\rho}}} \frac{1}{2\pi i} \int_L \frac{\Gamma\left(\frac{\gamma+1}{\rho} - \frac{s}{\rho}\right)\Gamma\left(\frac{1}{\beta-1} - \frac{\gamma+1}{\rho} + \frac{s}{\rho}\right)}{\Gamma\left(\frac{1}{\alpha-1}\right)\Gamma\left(\frac{1}{\beta-1}\right)}$$

$$\times \Gamma\left(\frac{s}{\delta}\right)\Gamma\left(\frac{1}{\alpha-1} - \frac{s}{\delta}\right)\left[\frac{z_2(\alpha-1)^{\frac{1}{\delta}}}{z_1(\beta-1)^{\frac{1}{\rho}}}\right]^{-s} ds$$

$$= \frac{c_1 c_2}{\delta\rho[z_2^\rho(\beta-1)]^{\frac{\gamma+1}{\rho}}\Gamma\left(\frac{1}{\alpha-1}\right)\Gamma\left(\frac{1}{\beta-1}\right)}$$

$$\times H_{2,2}^{2,2}\left[\frac{z_2(\alpha-1)^{\frac{1}{\delta}}}{z_1(\beta-1)^{\frac{1}{\rho}}} \middle| \begin{matrix} \left(1-\frac{\gamma+1}{\rho},\frac{1}{\rho}\right),\left(1-\frac{1}{\alpha-1},\frac{1}{\delta}\right) \\ \left(0,\frac{1}{\delta}\right),\left(\frac{1}{\beta-1}-\frac{\gamma+1}{\rho},\frac{1}{\rho}\right) \end{matrix}\right].$$

Therefore

$$I_4 = \int_0^\infty x^\gamma[1 + z_1^\delta(\alpha-1)x^\delta]^{-\frac{1}{\alpha-1}}[1 + z_2^\rho(\beta-1)x^\rho]^{-\frac{1}{\beta-1}} dx,$$

$$\alpha > 1, \beta > 1, \delta > 0, \rho > 0$$

$$= \frac{1}{\delta\rho[z_2^\rho(\beta-1)]^{\frac{\gamma+1}{\rho}}\Gamma\left(\frac{1}{\alpha-1}\right)\Gamma\left(\frac{1}{\beta-1}\right)}$$

$$\times H_{2,2}^{2,2}\left[\frac{z_2(\alpha-1)^{\frac{1}{\delta}}}{z_1(\beta-1)^{\frac{1}{\rho}}} \middle| \begin{matrix} \left(1-\frac{\gamma+1}{\rho},\frac{1}{\rho}\right),\left(1-\frac{1}{\alpha-1},\frac{1}{\delta}\right) \\ \left(0,\frac{1}{\delta}\right),\left(\frac{1}{\beta-1}-\frac{\gamma+1}{\rho},\frac{1}{\rho}\right) \end{matrix}\right].$$

Now by putting $y = \frac{1}{x}$ we can get an associated integral

$$I_4 = \int_0^\infty y^{-\gamma-2}[1 + z_1^\delta(\alpha-1)y^{-\delta}]^{-\frac{1}{\alpha-1}}[1 + z_2^\rho(\beta-1)y^{-\rho}]^{-\frac{1}{\beta-1}} dy.$$

Now, we can look at various special cases of $\lim_{\alpha\to 1}$ or $\lim_{\beta\to 1}$ or $\lim_{\alpha\to 1, \beta\to 1}$. These lead to some interesting special cases

$$I_{4.1} = \lim_{\alpha\to 1_+} I_4$$

$$= \int_0^\infty x^\gamma e^{-z_1^\delta x^\delta}[1 + z_2^\rho(\beta-1)x^\rho]^{-\frac{1}{\beta-1}} dx$$

$$= \frac{1}{\rho\delta[z_2^\rho(\beta-1)]^{\frac{\gamma+1}{\rho}}\Gamma\left(\frac{1}{\beta-1}\right)}H_{1,2}^{2,1}\left[\frac{z_2}{z_1(\beta-1)^{\frac{1}{\rho}}} \middle| \begin{matrix} \left(1-\frac{\gamma+1}{\rho},\frac{1}{\rho}\right) \\ \left(0,\frac{1}{\delta}\right),\left(\frac{1}{\beta-1}-\frac{\gamma+1}{\rho},\frac{1}{\rho}\right) \end{matrix}\right].$$

$$I_{4.2} = \lim_{\beta\to 1_+} I_4$$

$$= \int_0^\infty x^\gamma[1 + z_1^\delta(\alpha-1)x^\delta]^{-\frac{1}{\alpha-1}}e^{-z_2^\rho x^\rho} dx$$

$$= \frac{1}{\rho\delta z_2^{\gamma+1}\Gamma\left(\frac{1}{\alpha-1}\right)}H_{2,1}^{1,2}\left[\frac{z_2(\alpha-1)^{\frac{1}{\delta}}}{z_1} \middle| \begin{matrix} \left(1-\frac{\gamma+1}{\rho},\frac{1}{\rho}\right),\left(1-\frac{1}{\alpha-1},\frac{1}{\delta}\right) \\ \left(0,\frac{1}{\delta}\right) \end{matrix}\right],$$

$$I_{4.3} = \lim_{\alpha \to 1, \beta \to 1} I_4$$

$$= \int_0^\infty x^\gamma e^{-(z_1 x)^\delta - (z_2 x)^\rho} \mathrm{d}x$$

$$= \frac{1}{\rho \delta z_2^{\frac{\gamma+1}{\rho}}}$$

$$\times H_{1,1}^{1,1} \left[ \frac{z_2}{z_1} \middle| \begin{matrix} \left(1 - \frac{\gamma+1}{\rho}, \frac{1}{\rho}\right) \\ \left(0, \frac{1}{\delta}\right) \end{matrix} \right]$$

$$= \int_0^\infty x^{-\gamma-2} e^{-z_1^\delta x^{-\delta} - z_2^\rho x^{-\rho}} \mathrm{d}x.$$

When $\alpha < 1$ and $\beta < 1$ also we can obtain corresponding integrals, which are finite range integrals, by going through parallel procedure. In this case the limit of integration will be $0 < x < \max\{\epsilon_1, \epsilon_2\}$ where $\epsilon_1 = [z_1^\delta(\alpha - 1)]^{-\frac{1}{\delta}}$ and $\epsilon_2 = [z_2^\rho(\beta - 1)]^{-\frac{1}{\rho}}$.

Case of $\alpha < 1$, or $\beta < 1$.

When $\alpha < 1$, writing $\alpha - 1 = -(1 - \alpha)$ we can define the function

$$g_1(x) = x^\gamma [1 + z_1^\delta(\alpha - 1)x^\delta]^{-\frac{1}{\alpha-1}} = x^\gamma [1 - z_1^\delta(1 - \alpha)x^\delta]^{\frac{1}{1-\alpha}}, \alpha < 1 \tag{24}$$

for $[1 - z_1^\delta(1 - \alpha)x^\delta] > 0, \alpha < 1 \Rightarrow x < \dfrac{1}{z_1(1-\alpha)^{\frac{1}{\delta}}}$ and $g_1(x) = 0$ elsewhere. In this case the Mellin transform of $g_1(x)$ is the following:

$$h_1(s) = \int_0^\infty x^{s-1} g_1(x) \mathrm{d}x = \int_0^{\frac{1}{z_1(1-\alpha)^{\frac{1}{\delta}}}} x^{\gamma+s-1} [1 - z_1^\delta(1 - \alpha)x^\delta]^{\frac{1}{1-\alpha}} \mathrm{d}x \tag{25}$$

$$= \frac{1}{\delta[z_1(1-\alpha)^{\frac{1}{\delta}}]^{\gamma+s}} \frac{\Gamma(\frac{\gamma+s}{\delta})\Gamma(\frac{1}{1-\alpha} + 1)}{\Gamma(\frac{1}{1-\alpha} + 1 + \frac{\gamma+s}{\delta})}, \Re(\gamma + s) > 0, \alpha < 1, \delta > 0. \tag{26}$$

Then the Mellin transform of $f(z_2|z_1)$ for $\alpha < 1, \beta > 1$ is given by

$$M_{z_2|z_1}(s) = \frac{\Gamma(\frac{1}{1-\alpha} + 1)}{\delta \rho z_2^s z_1^{\gamma+s}(\beta - 1)^{\frac{s}{\rho}}(1 - \alpha)^{\frac{\gamma+s}{\delta}}} \frac{\Gamma(\frac{\gamma+s}{\delta})}{\Gamma(\frac{\gamma+s}{\delta} + \frac{1}{1-\alpha} + 1)} \frac{\Gamma(\frac{s}{\rho})\Gamma(\frac{1}{\beta-1} - \frac{s}{\rho})}{\Gamma(\frac{1}{\beta-1})}, \tag{27}$$

$$\Re(\gamma + s) > 0, \Re(s) > 0, \Re\left(\frac{1}{\beta - 1} - \frac{s}{\rho}\right) > 0.$$

Hence, the inverse Mellin transform for $\alpha < 1, \beta > 1$ is

$$f(z_2|z_1) = \frac{\Gamma(\frac{1}{1-\alpha} + 1)}{\delta \rho z_1^\gamma(1 - \alpha)^{\frac{\gamma}{\delta}}\Gamma(\frac{1}{\beta-1})}$$

$$\times H_{2,2}^{2,1} \left[ z_1 z_2(1 - \alpha)^{\frac{1}{\delta}}(\beta - 1)^{\frac{1}{\rho}} \middle| \begin{matrix} (1 - \frac{1}{\beta-1}, \frac{1}{\rho}), (1 + \frac{1}{1-\alpha} + \frac{\gamma}{\delta}, \frac{1}{\delta}) \\ (0, \frac{1}{\rho}), (\frac{\gamma}{\delta}, \frac{1}{\delta}) \end{matrix} \right], \tag{28}$$

$$\lim_{\beta \to 1} f(z_2|z_1) = \frac{\Gamma(\frac{1}{1-\alpha} + 1)}{\rho \delta z_1^\gamma(1 - \alpha)^{\frac{\gamma}{\delta}}} H_{1,2}^{2,0} \left[ z_1 z_2(1 - \alpha)^{\frac{1}{\delta}} \middle| \begin{matrix} (1 + \frac{1}{1-\alpha} + \frac{\gamma}{\delta}, \frac{1}{\delta}) \\ (0, \frac{1}{\delta}), (\frac{\gamma}{\delta}, \frac{1}{\delta}) \end{matrix} \right], \tag{29}$$

$$\lim_{\alpha \to 1} f(z_2|z_1) = \frac{1}{\rho \delta \Gamma(\frac{1}{\beta-1}) z_1^\gamma} H_{1,2}^{2,1} \left[ z_1 z_2(\beta - 1)^{\frac{1}{\rho}} \middle| \begin{matrix} (1 - \frac{1}{\beta-1}, \frac{1}{\rho}) \\ (0, \frac{1}{\rho}), (\frac{\gamma}{\delta}, \frac{1}{\delta}) \end{matrix} \right], \tag{30}$$

$$\lim_{\alpha \to 1, \beta \to 1} f(z_2|z_1) = \frac{1}{\rho \delta z_1^\gamma} H_{0,2}^{2,0} \left[ z_1 z_2 \middle|_{(0, \frac{1}{\rho}), (\frac{\gamma}{\delta}, \frac{1}{\delta})} \right]. \tag{31}$$

In $f(z_2|z_1)$ if $\beta < 1$ we may write $\beta - 1 = -(1 - \beta)$, and if we assume $[1 - z_2^\rho(1 - \beta)x^{-\rho}]^{\frac{1}{1-\beta}} > 0 \Rightarrow x > z_2(1 - \beta)^{\frac{1}{\rho}}$ then the corresponding integrals can also be evaluated as H-functions. But if $\alpha < 1$ and $\beta < 1$ then from the conditions

$$1 - z_1^\delta(1 - \alpha)x^\delta > 0 \Rightarrow x < \frac{1}{z_1(1-\alpha)^{\frac{1}{\delta}}} \text{ and } 1 - z_2^\rho(1-\beta)x^{-\rho} > 0 \Rightarrow x > z_2(1-\beta)^{\frac{1}{\rho}}$$

and the resulting integral may be zero. Hence, except this case of $\alpha < 1$ and $\beta < 1$ all other cases: $\alpha > 1, \beta > 1; \alpha < 1, \beta > 1; \alpha > 1, \beta < 1$ can be given meaningful interpretations as H-functions. Further, all these situations can be connected to practical problems. A few such practical situations will be briefly considered next.

## 2. Specific Applications

### 2.1. Krätzel Integral

For $\delta = 1, z_2^\rho = z, z_1 = 1$ in $f_3(z_2|z_1)$ gives the Krätzel integral

$$f_3(z_2|z_1) = \int_0^\infty x^{\gamma-1}e^{-x-zx^{-\rho}}dx \tag{32}$$

which was studied in detail by [4], see also [5]. Hence, $f_3$ can be considered as generalization of Krätzel integral. An additional property that can be seen from Krätzel integral as $f_3$ is that it can be written as a H-function of the type $H_{0,2}^{2,0}(\cdot)$. Hence, all the properties of H-function can now be made use of to study this integral further.

### 2.2. Inverse Gaussian Density in Statistics

Inverse Gaussian density is a popular density, which is used in many disciplines including stochastic processes. One form of the density is the following ([6], p. 33; [7]):

$$f(x) = c\,x^{-\frac{3}{2}}e^{-\frac{\lambda}{2}\left(\frac{x}{\mu^2}+\frac{1}{x}\right)}, \mu \neq 0, x > 0, \lambda > 0 \tag{33}$$

where $c = \pi^{-\frac{1}{2}}e^{\frac{\lambda}{|\mu|}}$. Comparing this with our case $f_3(z_1|z_2)$ we see that the inverse Gaussian density is the integrand in $f_3(z_1|z_2)$ for $\gamma = \frac{1}{2}, \rho = 1, z_2 = \frac{\lambda}{2}\left(\frac{1}{\mu^2}\right), \delta = 1, z_1 = \frac{\lambda}{2}$. Hence, $f_3$ can be used directly to evaluate the moments or Mellin transform in inverse Gaussian density.

### 2.3. Nuclear Reaction Rate Probability Integral in Astrophysics

A series of papers studied modifications to Maxwell–Boltzmann theory of stellar [8] and cosmological [9,10] nuclear reaction rates, a summary is given in [11]. The basic nuclear reaction rate probability integral that appears there is the following:

$$I_1 = \int_0^\infty x^{\gamma-1}e^{-ax-zx^{-\frac{1}{2}}}dx. \tag{34}$$

This is the case in the non-resonant case of nuclear reactions. Compare integral $I_1$ with $f_3(z_2|z_1)$. The reaction rate probability integral $I_1$ is $f_3(z_2|z_1)$ for $\delta = 1, \rho = \frac{1}{2}, z_2^{\frac{1}{2}} = z$. The basic integral $I_1$ is generalized in many different forms for resonant and non-resonant cases of reactions, depletion of high energy tail, and cut off of the high energy tail.

### 2.4. Tsallis' Non-Extensive Statistics and Beck–Cohen Superstatistics

Tsallis statistics is of the following form:

$$f_x(x) = c_1[1 + (\alpha - 1)x]^{-\frac{1}{\alpha - 1}}. \tag{35}$$

Compare $f_x(x)$ with the integrand in (1). For $z_2 = 0, \delta = 1, \gamma = 1$ the integrand in (1) agrees with Tsallis statistics $f_x(x)$ given in (32). The three different forms of Tsallis statistics are available from $f_x(x)$ for $\alpha > 1, \alpha < 1, \alpha \to 1$. The starting paper in non-extensive statistical mechanics may be seen from [12,13]. But the integrand in (1) with $z_2 = 0, z_1 = 1, \alpha > 1$ is the superstatistics of Beck and Cohen, see for example [14,15]. In statistical language, this superstatistics is the posterior density in a generalized gamma case when the scale parameter has a prior density belonging to the same class of generalized gamma density.

### 2.5. Pathway Model

Mathai (2005) [1] considered a rectangular matrix-variate function in the real case from where one can obtain almost all matrix-variate densities in current use in statistical disciplines. The corresponding version when the elements are in the complex domain is given in [16]. For the real scalar case the function is of the following form:

$$f(x) = c^*|x|^\gamma[1 - a(1 - \alpha)|x|^\delta]^{\frac{\eta}{1 - \alpha}} \tag{36}$$

for $-\infty < x < \infty, a > 0, \eta > 0, \delta > 0$ and $c^*$ is the normalizing constant. Here, $f(x)$ for $\alpha < 1$ stays in the generalized type-1 beta family when $[1 - a(1 - \alpha)|x|^\delta]^{\frac{\eta}{1 - \alpha}} > 0$. When $\alpha > 1$ the function switches into a generalized type-2 beta family, and when $\alpha \to 1$ it turns into a generalized gamma family of functions. Here, $\alpha$ behaves as a pathway parameter and hence the model is called a pathway model. Observe that the integrand in (1) is a product of two such pathway functions so that the corresponding integral is more versatile than a pathway model. Thus, for $z_2 = 0$ in (1) the integrand produces the pathway model of [1].

**Author Contributions:** Writing—original draft, A.M.M. and H.J.H.

**Funding:** This research received no external funding.

**Conflicts of Interest:** The authors declare no conflict of interest.

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
