# Peer review of "A Versatile Integral in Physics and Astronomy and Fox’s H-Function"

_axioms, doi:10.3390/axioms8040122_

Round 1

Reviewer 1 Report

This paper addresses a class of integrals that are found arise in variety of physical systems which arise from the Mathia model.

A nice part of the work is that the authors point out specific systems where their results apply to such as nuclear reaction rate, and Tsallis statistics.

It would be good if authors highlight better the results in abstract and summary which would make it easier for people doing literature searches to find these results if they want to apply them to their particular systems.

Author Response

We gratefully acknowledge the comments and suggestions of the Reviewer and have incorporated corrections in the revised manuscript.

Reviewer 2 Report

This paper contains some interesting ideas and useful in many areas and hence recommended for publication subject to revision by correcting the following errors: - Page 3, line 10: in the constant c the following two factors are missing  $\Gamma(1/{\alpha-1})$ $ \Gamma(1/{\beta-1})$ Please insert. - Page 4, end: Add the sentence "Also they fall into the category of Mellin convolution of a product involving two functions." - Page 5, line 19: Add "where $E(\dot)$ denotes the expected value of $(\dot)$." - Page 5,line last 2: In the numerator, $\Gamma$ is missing. Insert $\Gamma$; - Page 9,line 11 last: Change "(Bertulani, 2019)" to \(Bertulani, 2019a,b)" - Page 9, line last 2: Change Tsallis" to Tsallis' - Page 11: In the references of Bertulani put "a" for the firrst one and "b" for the second one and write as "2019a" and "2019b", respectively.   In my opinion with these changes the paper is recommended for publication in Axioms.

Author Response

We gratefully acknowledge the comments and suggestions of the Reviewer.

We have incorporated ALL corrections as outlined in the Reviewer's Report. These corrections are highlighted in yellow in the revised paper.